# RNase H enables efficient repair of R-loop induced DNA damage

**Jeremy D Amon, Douglas Koshland\***

Department of Molecular and Cell Biology, University of California, Berkeley, Berkeley, United States

**Abstract** R-loops, three-stranded structures that form when transcripts hybridize to chromosomal DNA, are potent agents of genome instability. This instability has been explained by the ability of R-loops to induce DNA damage. Here, we show that persistent R-loops also compromise DNA repair. Depleting endogenous RNase H activity impairs R-loop removal in *Saccharomyces cerevisiae*, causing DNA damage that occurs preferentially in the repetitive ribosomal DNA locus (rDNA). We analyzed the repair kinetics of this damage and identified mutants that modulate repair. We present a model that the persistence of R-loops at sites of DNA damage induces repair by break-induced replication (BIR). This R-loop induced BIR is particularly susceptible to the formation of lethal repair intermediates at the rDNA because of a barrier imposed by RNA polymerase I.

*For correspondence: koshland@berkeley.edu

**Competing interests:** The authors declare that no competing interests exist.

## Introduction

R-loops are structures that form when RNA invades double-stranded DNA and hybridizes to complementary genomic sequences (*Gaillard and Aguilera, 2016*). R-loops can form spontaneously across many genomic loci, but the activity of two endogenous RNases H prevents their accumulation and persistence (*Cerritelli and Crouch, 2009*). RNase H1 and H2 are highly conserved ribonucleases with the ability to degrade the RNA moiety of a DNA:RNA hybrid. Disrupting the activity of the two enzymes (*rnh1Δ rnh201Δ* in *Saccharomyces cerevisiae*) has been a useful tool for increasing the persistence of DNA:RNA hybrids and studying the effects of hybrid-induced instability. Indeed, efforts to map R-loops genome-wide have shown that in the absence of RNase H activity, the levels of hybrids formed at susceptible loci increase dramatically (*Wahba et al., 2016*). This increase in hybrids is associated with increased rates of genome instability that include loss of heterozygosity (LOH) events, loss of entire chromosomes, and recombination at the ribosomal locus (*Wahba et al., 2011*; *O'Connell et al., 2015*). The RNases H have therefore been implicated as important protectors of genome stability.

The ribosomal locus (rDNA) appears to be particularly prone to R-loops. Approximately 60% of all transcription in *S. cerevisiae* is devoted to producing ribosomal RNA from about 150 repeated units located in a clustered region on chromosome XII (*Warner, 1999*). These repeats, at 9.1 kb each, make up about 10% of the budding yeast genome. Accordingly, almost 50% of all R-loops map to the rDNA (*Wahba et al., 2016*). R-loops found at the rDNA are associated with increased rates of recombination (*Wahba et al., 2011*, *2013*), RNA polymerase pileups (*El Hage et al., 2010*), and stalled replication forks (*Stuckey et al., 2015*).

A growing body of evidence has attributed various biological roles to R-loops, including modifying gene expression (*Ginno et al., 2012*; *Sun et al., 2013*), terminating transcription (*Skourti-Stathaki et al., 2011*, *2014*), driving sequence mutation (*Gómez-González and Aguilera, 2007*), and inducing changes in genome structure (*Li and Manley, 2005*; *Ruiz et al., 2011*). However, the mechanisms of R-loop induced genome instability remain elusive. Most studies on the mechanisms

of hybrid-induced instability have been 'damage-centric,' investigating how R-loops are converted to mutations, single-stranded nicks, and double-stranded breaks (DSBs) (*Aguilera and García-Muse, 2012*). Current models focus on the involvement of active replication forks that stall or collapse upon encountering the aberrant structure. While this remains an area of active research, we note that any instability event is the result of a complex interplay between the initial damage event and the repair processes that follow. Phenotypes that involve the loss of genetic information (terminal deletions, certain LOH events) imply both that damage occurred and that repair processes failed to accurately maintain the genome. Few studies have investigated how R-loop induced damage is repaired, and it remains possible that defects in repair contribute to instability. This possibility raises several questions. First, do genomic changes induced by R-loops reflect increases in damage events, failures of repair, or both? Second, are specific pathways involved in the repair of R-loop induced damage, and if so, what are they?

To begin to answer these questions, we turned to the Rad52-GFP foci system in *S. cerevisiae*. Rad52 is required in almost all homologous recombination (HR) pathways, and in yeast forms bright foci upon induction of DNA damage (*Lisby et al., 2001*). Most foci appear in the S/G2-M phases of the cell cycle and have a moderate rate of repair – almost all spontaneous Rad52-GFP foci are resolved within 40 min (*Lisby et al., 2003*). Consistent with phenotypes of increased genomic instability, *rnh1Δ rnh201Δ* mutants display an increase in Rad52-GFP foci. A large fraction of these foci appear to co-localize with the nucleolus and form in a window between late S and mid-M (*Stuckey et al., 2015*). Here, by monitoring the persistence of Rad52 foci across the cell cycle in RNase H mutants, we implicate DNA:RNA hybrids in the disruption of DNA repair. We show that topoisomerase I works at the rDNA to prevent these disruptions from becoming lethal events. Furthermore, we identify a new role for the RNases H in preventing break-induced replication (BIR) from repairing R-loop induced DNA damage.

## Results

### The presence of either RNase H1 or H2 prevents the accumulation of DNA damage in G2-M

To better understand the mechanisms by which DNA:RNA hybrids contribute to genome instability, we began by characterizing DNA damage in exponentially dividing wild-type, *rnh1Δ*, *rnh201Δ*, and *rnh1Δ rnh201Δ* budding yeast cells. Using Rad52-GFP foci as a marker for DNA damage, we observed that 27% of *rnh1Δ rnh201Δ* cells had foci, a ten-fold increase over wild-type, *rnh1Δ*, and *rnh201Δ* cells (*Figure 1A*). Consistent with the notion that persistent DNA damage uniquely affects the double mutants, the growth of the double mutant, but not either of the single mutants, was dramatically impaired by the deletion of *RAD52* (*Figure 1B*). Previous characterization of the double mutant also reported elevated foci and Rad52-dependent growth (*Stuckey et al., 2015*; *Lazzaro et al., 2012*). Thus, by measures of Rad52-GFP foci and Rad52-dependent growth, cells lacking RNase H1 and H2 had a larger fraction of persistent R-loop induced damage than wild-type cells or cells lacking only one of the RNases H. This persistent damage could have arisen from increased R-loop induced damage and/or an inability to efficiently repair that damage.

To further characterize the DNA damage response in *rnh1Δ rnh201Δ* cells, we asked whether this damage accumulated within a specific window of the cell cycle. We arrested *rnh1Δ rnh201Δ* cells in G1 using the mating pheromone alpha factor and released them into nocodazole, allowing them to proceed synchronously through the cell cycle until they arrested in mid-M phase at the spindle checkpoint (*Figure 1C*, *Figure 1—figure supplement 1*). During this cell-cycle progression, aliquots of cells were removed and fixed to assess Rad52-GFP foci accumulation. Cell-cycle stage was determined by measuring DNA content using flow cytometry (*Figure 1—figure supplement 1*). The fraction of cells with Rad52-GFP foci remained around 10 to 15 percent through S-phase. Additional foci appeared at the S/G2-M boundary and accumulated to around 50 percent, as reported previously.

The failure to observe accumulating foci early in the cell cycle was not a limitation of the system, as an identical analysis of a single cell cycle of *sin3Δ* cells, which also accumulate hybrids, revealed an increase in focus formation during S-phase (*Figure 1—figure supplement 2A and B*). The increase in foci in *rnh1Δ rnh201Δ* cells did not appear to be due to a cell-cycle dependent increase in hybrid formation, as cytological staining revealed similar levels of R-loops in cells staged in G1, S,

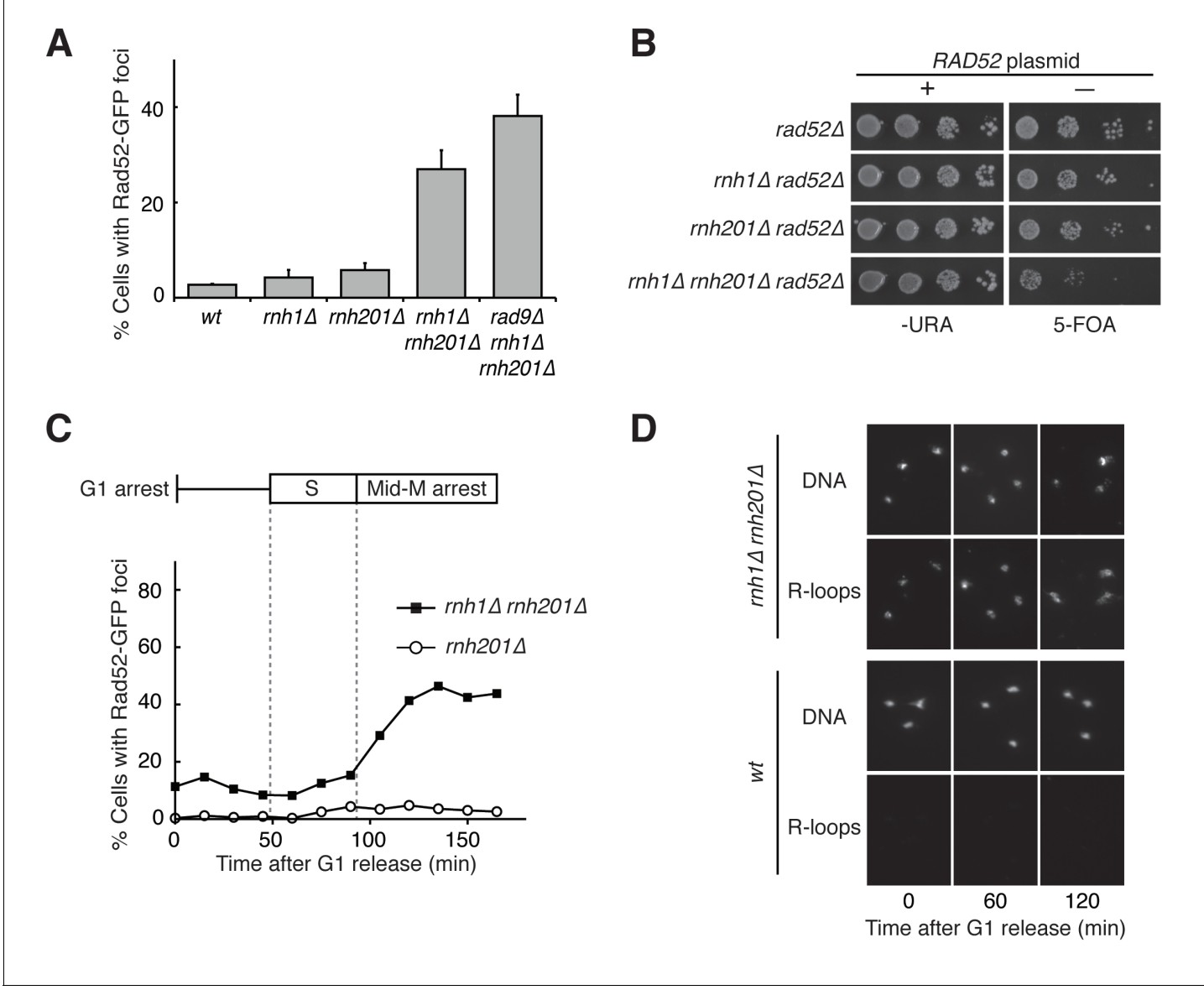

**Figure 1.** Cells lacking both RNases H accumulate DNA damage in G2-M. (**A**) Assessment of Rad52-GFP foci in RNase H mutants. Asynchronously dividing cells were scored for the presence of one or more Rad52-GFP focus. Bars represent mean +/- standard deviation ($n = 3$; 300 cells scored per replicate). (**B**) Assessment of Rad52 requirement in RNase H mutants. Cells carrying a plasmid expressing *RAD52* and *URA3* were plated onto media lacking uracil (-URA, selects for plasmid) or media containing 5-floroorotic acid (5-FOA, selects for plasmid loss). 10-fold serial dilutions are shown. (**C**) Cell cycle profile of Rad52-GFP foci in RNase H mutants. Synchronously dividing cells were scored for the presence of Rad52-GFP foci. Cells arrested in G1 using alpha factor were washed and released into nocodazole. Samples were taken at 15 min intervals and 300 cells per time point were scored for Rad52-GFP foci. Cell cycle phase is determined by flow cytometry (*Figure 1—figure supplement 1*). (**D**) Cell cycle profile of DNA:RNA hybrids in RNase H mutants. Representative images of chromosome spreads of *rnh1Δ rnh201Δ* and *wild-type* cells are shown. Spreads are stained for DNA content (DAPI) or immunostained for DNA:RNA hybrids (R-loops) using the S9.6 antibody and a fluorescent-conjugated secondary.

The following figure supplements are available for figure 1:

**Figure supplement 1.** Flow cytometry of *rnh201Δ* and *rnh1Δ rnh201Δ* cells released from alpha factor into nocodazole.

**Figure supplement 2.** Deleting *SIN3* causes increased foci in S phase.

**Figure supplement 3.** Quantification of *Figure 1D*.

and M, with around 95% of all nuclei staining for presence of R-loops (*Figure 1D*, *Figure 1—figure supplement 3*). Therefore, the increase in damage during the S/G2-M window in *rnh1Δ rnh201Δ* cells likely occurred because hybrids were more efficiently converted to damage. The mechanism of the hybrid-induced damage observed here could have involved collisions with late-firing replication forks (reviewed in [*Gaillard and Aguilera, 2016*]). Alternatively, the repair of hybrid-induced damage became impaired.

The presence of DNA damage such as DSBs leads to a Rad9-dependent cell-cycle checkpoint that delays entry into anaphase (*Weinert and Hartwell, 1988*). We found that the fraction of cycling cells in G2-M, defined as a large-budded morphology with an undivided nucleus, was two-fold higher in *rnh1Δ rnh201Δ* cells than wild-type or either RNase H single mutant. This fraction was reduced by deletion of *RAD9* (*Figure 2A*). Deletion of *RAD9* did not decrease the level of Rad52-GFP foci in *rnh1Δ rnh201Δ* cells, indicating that focus formation was not dependent on the checkpoint (*Figure 1A*).

To assess the kinetics of foci persistence in *rnh1Δ rnh201Δ* cells, we arrested cultures in S-phase using hydroxyurea and released them into alpha factor, allowing them to proceed through M-phase and arrest in the following G1 (*Figure 2B* and *Figure 2—figure supplement 1*). After the expected increase of Rad52-GFP foci upon the completion of S-phase, we observed a gradual disappearance of foci. Throughout the time-course, the vast majority of cells that retained foci were arrested pre-anaphase, indicating that most cells delayed progression into anaphase until the damage was repaired (*Figure 2B*). For example, after 330 min, the bulk of cells had reached G1 (*Figure 2C*) and the fraction of cells with Rad52-GFP foci had dropped to 20 percent. Of the cells that retained foci, 77 percent remained arrested in G2-M before anaphase. The slow disappearance of foci and progression into anaphase raised the possibility that hybrid-induced damage might be difficult to repair in a subset of the double mutant cells.

## Topoisomerase-1 cooperates with RNase H1 and H2 to prevent the accumulation of DNA damage in G2-M

To improve our ability to interrogate the unusual DNA damage in *rnh1Δ rnh201Δ* cells, we sought to strengthen the damage phenotype. A number of observations suggested that alleles of *TOP1*, which encodes the major topoisomerase I in yeast, might be good candidates for doing so. Top1 is thought to clear or prevent R-loops and stalled RNA polymerase I (RNA pol I) complexes at the ribosomal locus by resolving supercoiling (*El Hage et al., 2010*; *Drolet et al., 1995*; *Massé and Drolet, 1999*). A potential synergistic relationship between Top1 and the RNases H came from the observation that while cells with either the *top1Δ* mutation or the *rnh1Δ rnh201Δ* mutations are viable, the *top1Δ rnh1Δ rnh201Δ* mutant is inviable (*El Hage et al., 2010*). Furthermore, treatment of *rnh1Δ rnh201Δ* cells with the Top1 inhibitor camptothecin led to increased Rad52-GFP foci that co-localized with the nucleolus (*Stuckey et al., 2015*). Encouraged by these results, we used the auxin-inducible degron (AID) system to create a conditional *TOP1-AID* allele in wild-type, the two single RNase H mutants, and the double RNase H mutant. We then reassessed viability and DNA damage phenotypes.

Consistent with published results, *rnh1Δ rnh201Δ TOP1-AID* cells failed to grow when treated with auxin (*Figure 3A*). In contrast, *TOP1-AID*, *rnh1Δ TOP1-AID*, and *rnh201Δ TOP1-AID* mutants grew well. Thus, the synergistic lethality occurred only when both RNases H and Top1 were inactivated. Similarly, when exponential cultures of these strains were treated with auxin for four hours, Rad52-GFP foci did not increase in *TOP1-AID*, *rnh1Δ TOP1-AID* or *rnh201Δ TOP1-AID* mutants (*Figure 3B*). However, foci nearly doubled in the *rnh1Δ rnh201Δ TOP1-AID* cells compared to an untreated control, such that a large majority of *rnh1Δ rnh201Δ TOP1-AID* cells (85%) had foci. Furthermore, after four hours of treatment with auxin, over 98 percent of *rnh1Δ rnh201Δ TOP1-AID* cells were arrested pre-anaphase at the G2-M checkpoint (*Figure 3C and D*). This arrest reflected an exacerbation of the cell cycle delay observed in the *rnh1Δ rnh201Δ* strain and in *rnh1Δ rnh201Δ TOP1-AID* cells left untreated with auxin (*Figures 2A* and *3C*). As with *rnh1Δ rnh201Δ* cells, the cell-cycle arrest of the *rnh1Δ rnh201Δ TOP1-AID* was Rad9 dependent – deletion of *RAD9* resulted in cells that proceeded into the following G1. Importantly, deletion of *RAD9* did not restore viability to *rnh1Δ rnh201Δ TOP1-AID* cells treated with auxin. This result suggested that the inviability of the triple mutant was not simply due to the constitutive activation of the checkpoint but rather to the presence of irreparable damage.

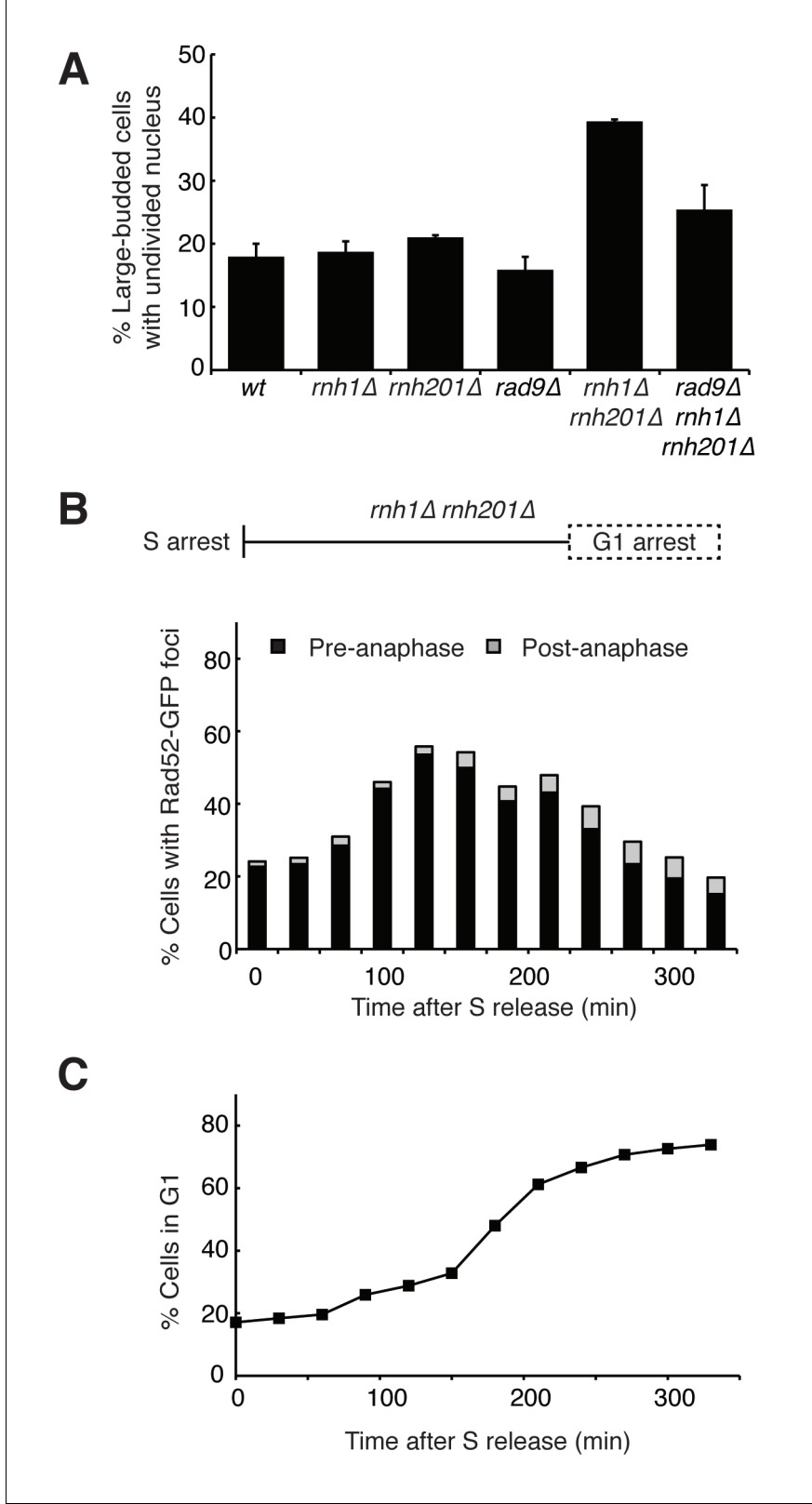

**Figure 2.** Cells with hybrid-induced DNA damage are slow to repair. (**A**) Assessment of cell-cycle delay in RNase H mutants. Asynchronously dividing cells were scored on the basis of their bud size and nuclear morphology. The percentage of cells with large buds and an undivided nucleus (single DAPI mass) are shown. Bars represent mean

*Figure 2 continued*

+/- standard deviation (*n* = 3, 100 cells scored per replicate) (**B**) Cell cycle profile of Rad52-GFP foci in dividing cells. *rnh1Δ rnh201Δ* cells were arrested in S-phase using hydroxyurea, washed, and released into alpha factor. Samples were taken at 30 min intervals and 300 cells per time point were scored for Rad52-GFP foci. If a cell had a Rad52-GFP focus, it was further scored for cell cycle phase. Cells with undivided nuclei (single DAPI mass) are labeled 'pre-anaphase,' while those that had undergone nuclear division (two DAPI masses or G1 arrested) are labeled 'post-anaphase.' (**C**) Cell cycle stage of dividing *rnh1Δ rnh201Δ* cells. Cells from (**B**) were subjected to flow cytometry (*Figure 2—figure supplement 1*) and quantified. The percentage of cells with 1C DNA content is shown.

The following figure supplement is available for figure 2:

**Figure supplement 1.** Flow cytometry of *rnh1Δ rnh201Δ* cells released from hydroxyurea into alpha factor.

---

A striking feature of the Rad52-GFP foci in the *rnh1Δ rnh201Δ* double mutant was that they accumulated in a window that began at the boundary between S and G2-M. Therefore, we tested whether the enhanced focus formation in the *rnh1Δ rnh201Δ TOP1-AID* cells also occurred in this window (*Figure 4A*). A culture of the *rnh1Δ rnh201Δ TOP1-AID* triple mutant was arrested in G1 with alpha factor and treated with auxin to deplete Top1-AID (*Figure 4—figure supplement 1A and C*). Cells were released from G1 into media containing auxin and nocodazole, to perpetuate Top1-AID depletion and induce subsequent arrest in mid-M (*Figure 4—figure supplement 1A*). Aliquots were removed as cells progressed from G1 to mid-M arrest and assessed for Rad52-GFP foci. As a control, a second culture was subjected to the same regime without auxin.

The enhanced foci in the triple mutant exhibited the same kinetics of accumulation as the double mutant (*Figure 4A*). The fraction of triple mutant cells with Rad52-GFP foci in both cultures remained around 15 to 20 percent until the end of bulk S-phase, similar to the *rnh1Δ rnh201Δ* double mutant. At subsequent time points, the auxin-free triple mutant culture mimicked the double mutant, as foci rose to about 45 percent in G2-M. The fraction of Rad52-GFP foci in the triple mutant cells treated with auxin also rose in G2-M but to a higher value of about 75 percent. As we suggested for double mutants, the accumulation of DNA damage after bulk S-phase was consistent with DSBs being formed by impaired late-firing replication forks, such as those seen at the rDNA. Taken together, the triple mutant exhibited qualitatively similar but quantitatively greater cell-cycle and DNA damage defects relative to the double mutant, indicating that Top1 depletion enhanced the DNA damage phenotype caused by loss of RNase H activity.

The ability to conditionally inactivate Top1-AID allowed us to address the role of Top1 activity in the cell-cycle dependent appearance of Rad52-GFP foci and the connection between focus formation and lethality. Cultures of *rnh1Δ rnh201Δ TOP1-AID* cells were arrested in mid-M with nocodazole and then treated with auxin. Top1-AID was depleted within 30 min of addition of auxin (*Figure 4—figure supplement 1B and C*). The fraction of cells with Rad52-GFP foci climbed to about 70 percent (*Figure 4B*). This result suggested that Top1 activity was required after the completion of bulk S-phase to prevent focus formation.

To address whether Rad52-GFP foci induced by Top1 depletion were correlated with lethality, asynchronously dividing *rnh1Δ rnh201Δ TOP1-AID* cells were transiently treated with auxin for four hours, washed with fresh media, and then plated onto nonselective plates. The fraction of cells that survived, relative to an untreated control, was around 16 percent, similar to the percentage of cells that did not have Rad52 foci when given the same treatment (*Figure 4C*). This correlation suggested that the persistent foci in the *rnh1Δ rnh201Δ TOP1-AID* cells represented lethal DNA damage that arose from inactivation of Top1 activity in G2-M.

To test this hypothesis further, we asked whether the appearance of foci was temporally correlated with inviability. The *rnh1Δ rnh201Δ TOP1-AID* cells were first arrested in S- or mid-M phase with hydroxyurea or nocodazole, respectively. The arrested cells were treated with auxin to deplete Top1-AID activity. These cells were plated for viability on media lacking auxin, allowing the restoration of Top1-AID activity (*Figure 4C*). Transiently depleting Top1 in S-phase, in which Rad52-GFP foci levels remain unchanged (18%), did not lead to loss of viability. However, transiently depleting Top1 in mid-M phase, which led to elevated foci levels (75%), also led to a dramatic increase in

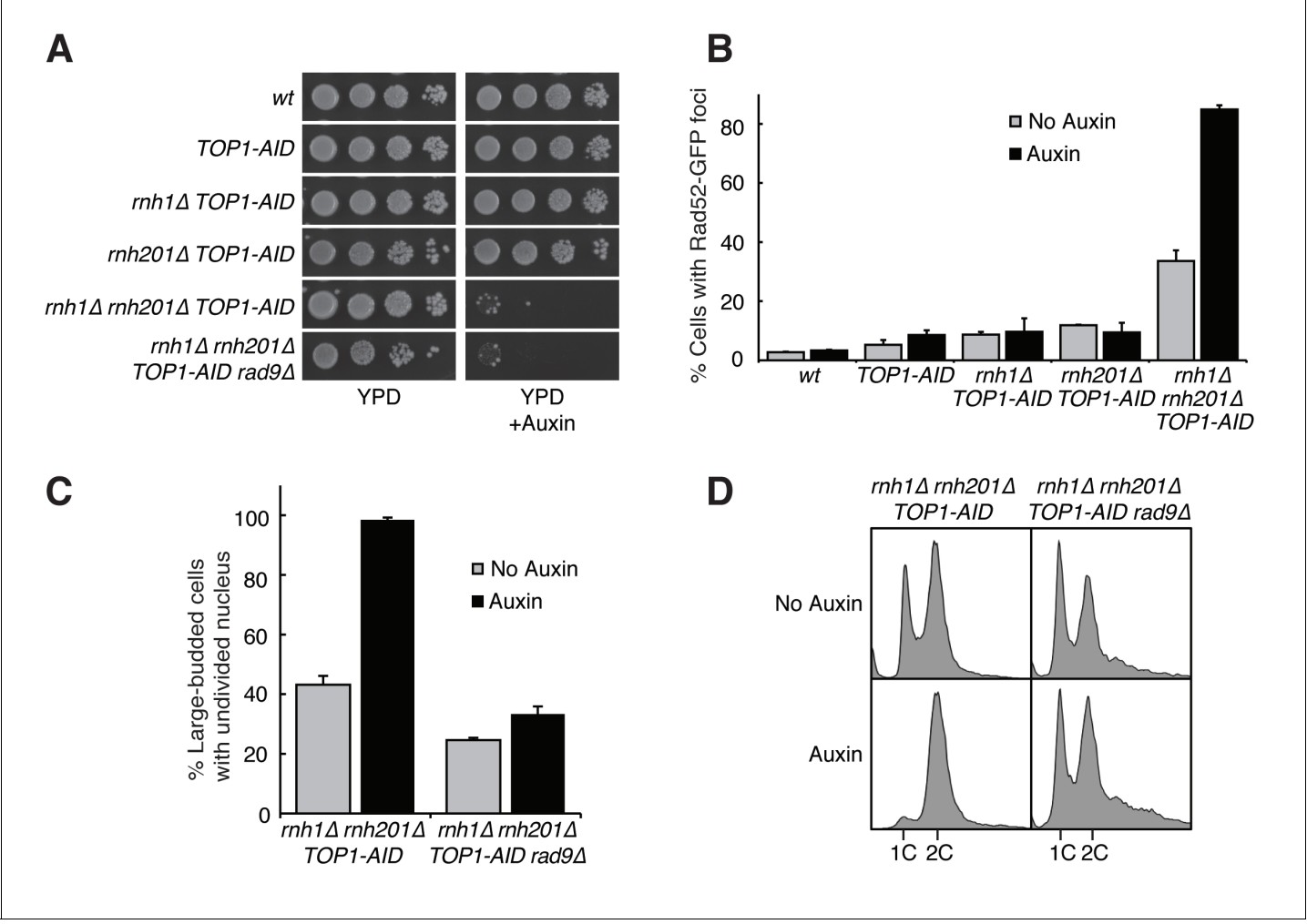

**Figure 3.** Depleting topoisomerase I exacerbates *rnh1Δ rnh201Δ* phenotypes. (**A**) Assessment of Top1 depletion on viability of RNase H mutants. 10-fold serial dilutions of saturated cultures were plated onto rich media (YPD) or media containing auxin (YPD +Auxin). (**B**) Assessment of Top1 depletion on Rad52-GFP foci in RNase H mutants. Cultures were grown at 23 degrees and treated with auxin for four hours. Cells were then scored for presence of Rad52-GFP foci. Bars represent mean +/- standard deviation (n = 3, 300 cells scored per replicate). (**C**) Depleting Top1 leads to robust Rad9-dependent cell cycle arrest. Logarithmically dividing cells were treated with auxin for four hours then scored for bud size and nuclear morphology. The percentage of cells with large buds and undivided nuclei (single DAPI mass) is shown. Bars represent mean +/- standard deviation (n = 3, 100 cells scored per replicate). (**D**) Cells from (**C**) were subjected to flow cytometry.

lethality. These results suggested that depletion of Top1 activity in G2-M in cells lacking the RNases H led to irreparable DNA damage in the vast majority of cells.

## Screen for suppressors of hybrid-induced lethality

Taken together, our data suggested that *rnh1Δ rnh201Δ* cells depleted of Top1 initiated homologous recombination, but ultimately could not repair R-loop induced damage. This led us to question which HR pathway was being initiated and why the damage was not repairable. The lethality of *rnh1Δ rnh201Δ top1Δ* cells provided a powerful genetic tool to do so. We reasoned that suppressor mutations that confered viability to a strain lacking the RNases H and Top1 could either prevent damage from occurring or allow that damage to become repairable. These suppressor mutations could inform us about the processes that convert R-loops to DNA damage or the mechanisms by which R-loop mediated damage is repaired. To isolate these suppressors, we generated independent cultures of an *rnh1Δ rnh201Δ top1Δ* strain that contained a plasmid carrying the *RNH1* and *URA3* genes. Plating these cultures on 5-fluoroorotic acid (5-FOA) selected for cells that had lost the

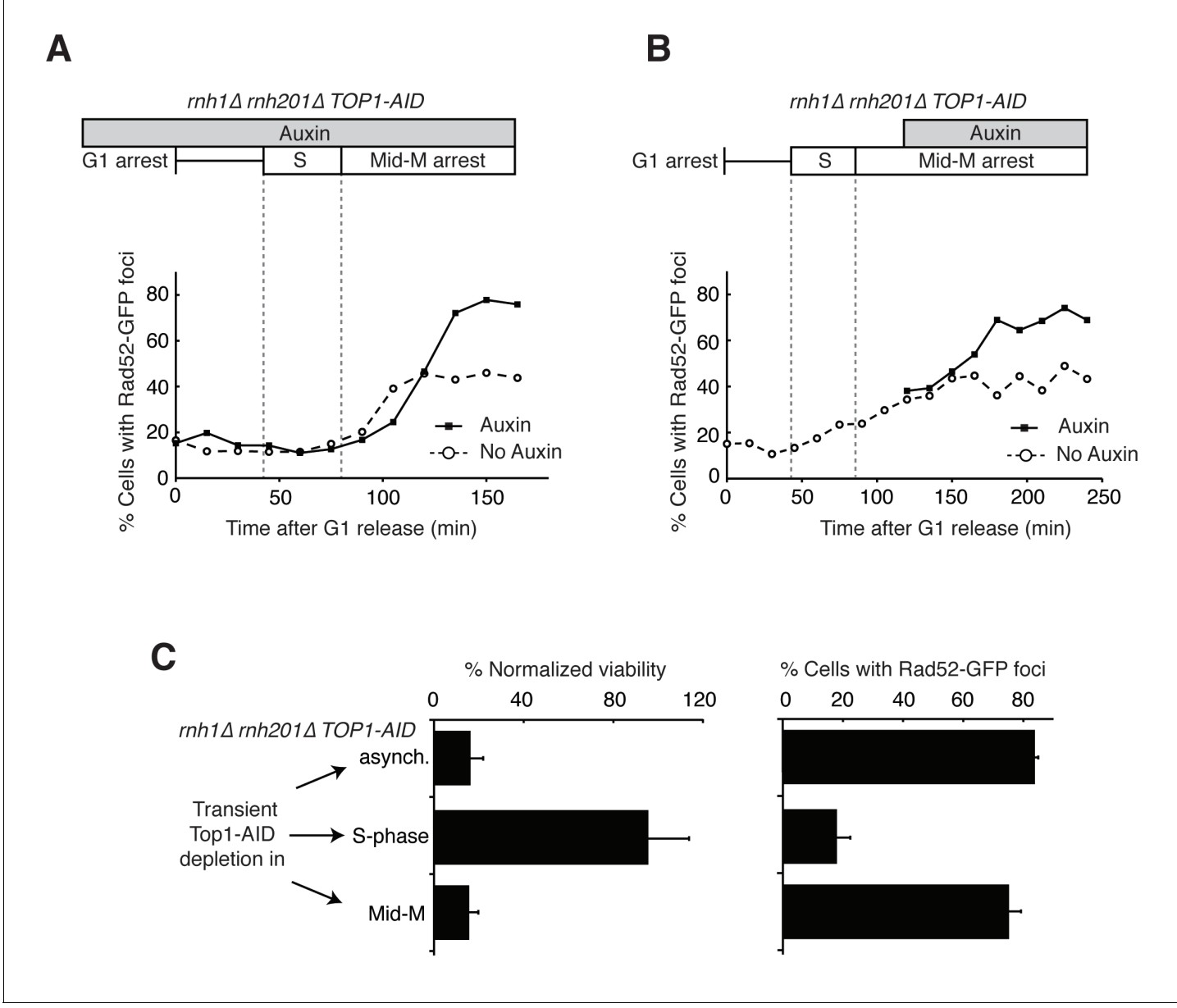

**Figure 4.** Depleting topoisomerase I causes lethal DNA damage in G2-M. (**A**) Depleting Top1 in *rnh1Δ rnh201Δ* cells shows similar onset of Rad52-GFP at the S/G2-M border. Cultures of *rnh1Δ rnh201Δ TOP1-AID* cells were arrested in G1 using alpha factor, treated with auxin for 2 hr, then released into media containing nocodazole and auxin. Samples were taken at 15 min intervals and 300 cells per time point were scored for Rad52-GFP foci. (**B**) Depleting Top1 in *rnh1Δ rnh201Δ* cells after completion of S-phase causes accumulation of Rad52-GFP foci. Cultures of *rnh1Δ rnh201Δ TOP1-AID* cells were released from alpha factor into nocodazole. Once cells had completed S-phase, auxin was added. Samples were taken at 30 min intervals and 300 cells per time point were scored for Rad52-GFP foci. (**C**) Cultures of *rnh1Δ rnh201Δ TOP1-AID* cells were allowed to divide asynchronously, arrested in S-phase using hydroxyurea, or arrested in Mid-M phase using nocodazole. Once cells were arrested, auxin was added for four hours. *Left* – Cells were washed and plated on YPD for recovery. Viability was measured by normalizing colony-forming units from auxin-treated cells to untreated cells. Bars represent mean +/- standard deviation (*n = 4*). *Right* – Cells were fixed and scored for Rad52-GFP foci. Bars represent mean +/- standard deviation (*n = 3*, 300 cells scored per replicate).

The following figure supplement is available for figure 4:

**Figure supplement 1.** Details on *rnh1Δ rnh201Δ TOP1-AID* cells released from alpha factor into nocodazole.

plasmid and thus carried a suppressor mutation that allowed them to divide despite their *rnh1Δ rnh201Δ top1Δ* genotype (*Figure 5A*). DNA sequencing of the suppressor strains identified novel mutations in the Pif1 helicase and in RNA polymerase I. The analysis of these suppressors described below provided new insights into the HR pathway initiated by hybrid-induced damage and the mechanisms by which DNA repair fails.

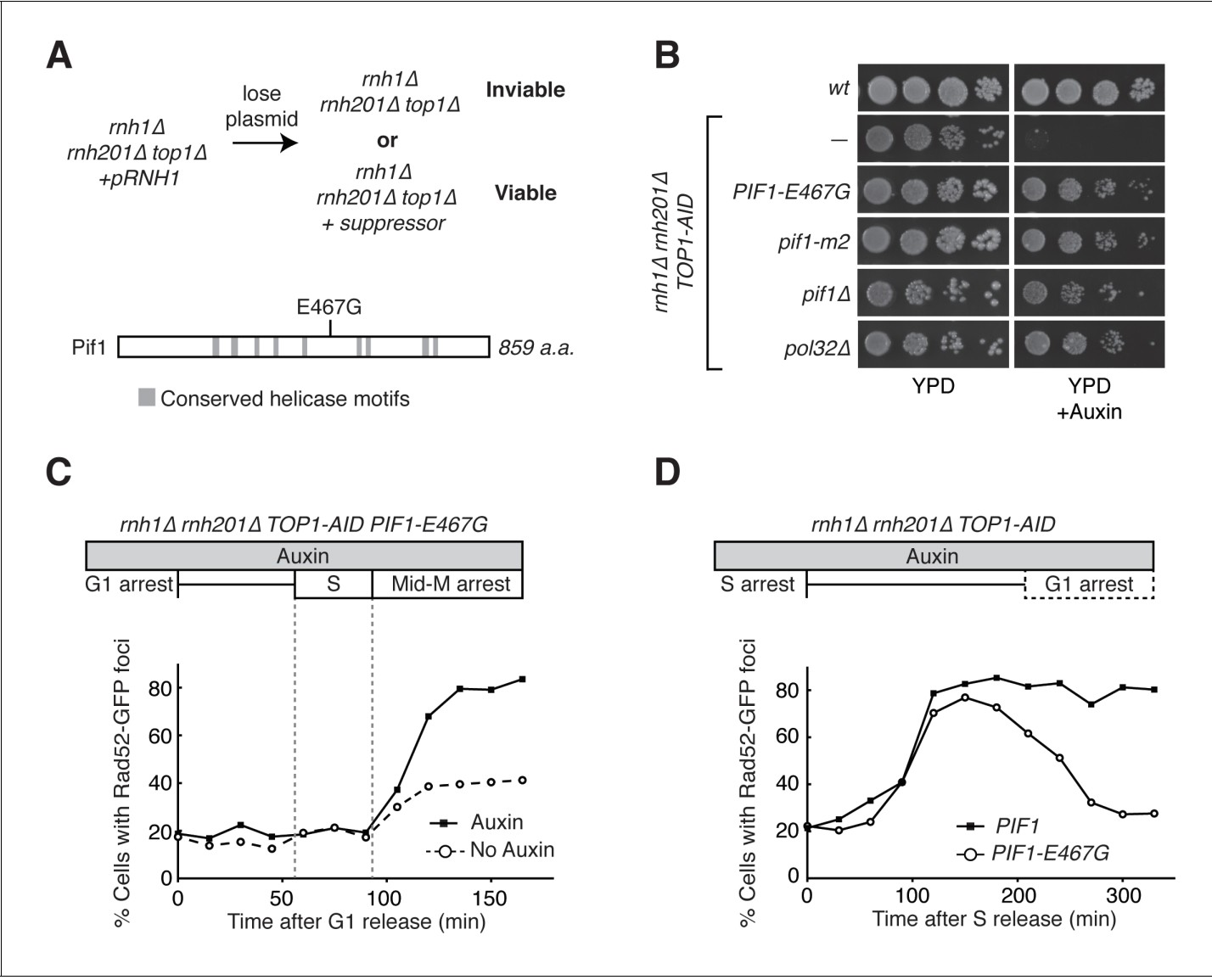

**Figure 5.** *Pif1-E467G* allows for repair of R-loop induced damage. (**A**) *Top*: schematic of genetic screen for suppressors of hybrid-induced lethality. Cells for the screen were *rnh1Δ rnh201Δ top1Δ* and carried a plasmid expressing *RNH1* and *URA3*. Cultures were grown in non-selective media and plated onto 5-FOA to select for cells that had lost the plasmid and therefore gained suppressor mutations of *rnh1Δ rnh201Δ top1Δ* lethality. *Bottom*: schematic of Pif1 showing location of E467 relative to evolutionarily conserved SFI helicase motifs and motifs conserved between Pif1 and RecD, as previously published (***Boulé and Zakian, 2006***). (**B**) Mutations in *PIF1* and *POL32* suppress auxin sensitivity of *rnh1Δ rnh201Δ TOP1-AID* cells. 10-fold serial dilutions of saturated cultures were plated onto YPD or YPD with auxin. (**C**) Pif1-E467G does not change accumulation of Rad52-GFP foci. Experiment in ***Figure 4A*** was repeated on *rnh1Δ rnh201Δ TOP1-AID PIF1-E467G* cells. (**D**) Pif1-E467G allows for repair of Rad52-GFP foci. Experiment in ***Figure 2B*** was repeated on *rnh1Δ rnh201Δ TOP1-AID* cells in the presence of auxin with or without *PIF1-E467G*.

The following figure supplement is available for figure 5:

**Figure supplement 1.** Flow cytometry on *rnh1Δ rnh201Δ TOP1-AID PIF1-E467G* cells.

## The processivity step in break-induced replication is responsible for the inability to repair hybrid-induced DNA damage

Our screen identified *PIF1-E467G*. This allele suppressed auxin sensitivity when introduced into *rnh1Δ rnh201Δ TOP1-AID* cells, indicating its responsibility for the suppression of lethality in *rnh1Δ rnh201Δ top1Δ* cells (*Figure 5B*). This allele was not found in any previously described domains of Pif1 (*Figure 5A*), prompting us to assess the ability of well-characterized recessive *PIF1* alleles to suppress the lethality of *rnh1Δ rnh201Δ top1Δ* genotype (*Boulé and Zakian, 2006*). When introduced into *rnh1Δ rnh201Δ TOP1-AID*, both *pif1Δ* and *pif1-m2*, an allele that maintains mitochondrial but not nuclear functions of Pif1 (*Schulz and Zakian, 1994*), were able to suppress the auxin sensitivity of *rnh1Δ rnh201Δ TOP1-AID* cells. We concluded that *PIF1-E467G* likely inactivated a nuclear activity that contributed to hybrid-induced lethality (*Figure 5B*).

To determine whether *PIF1-E467G* prevented hybrid-induced DNA damage in *rnh1Δ rnh201Δ TOP1-AID* cells or allowed for its repair, we monitored the appearance and disappearance of Rad52-GFP foci in synchronously dividing cells. A culture of *rnh1Δ rnh201Δ TOP1-AID PIF1-E467G* cells was arrested in G1 (alpha factor) and treated with auxin to deplete Top1-AID. The culture was switched into media containing nocodazole and auxin to perpetuate Top1-AID depletion and allow progression through the cell cycle until arrest in mid-M phase (*Figure 5C*, *Figure 5—figure supplement 1A*). The pattern of appearance of Rad52-GFP foci in this culture showed a strong similarity to the parent strain expressing wild-type Pif1, with no increase in foci until the completion of bulk S-phase. The fraction of cells containing Rad52-GFP foci then rose to 80 percent with auxin treatment and 40 percent without. These results suggested that the *PIF1-E467G* allele did not prevent hybrid induced DNA damage.

We next compared the appearance and disappearance of Rad52-GFP foci in *rnh1Δ rnh201Δ TOP1-AID* and *rnh1Δ rnh201Δ TOP1-AID PIF1-E467G* strains as they progressed between S phase to the subsequent G1. Cultures of these cells were arrested in hydroxyurea to induce S-phase arrest and then treated with auxin to induce Top1-AID depletion. The cultures were switched into media with auxin and alpha factor to perpetuate Top1-AID depletion and allow progression through mitosis to the next G1 (*Figure 5D*, *Figure 5—figure supplement 1B*). In contrast to *rnh1Δ rnh201Δ TOP1-AID* cells, which maintained 85% Rad52-GFP foci and never proceeded through anaphase, cells with *PIF1-E467G* gradually resolved most of their foci as they completed mitosis. We therefore concluded that *PIF1-E467G* promoted the repair of hybrid-induced damage.

These observations led us to question the role of break-induced replication (BIR) in hybrid-mediated instability. BIR is an HR-dependent repair process that can occur in G2-M when only one end of a DSB is available for recombination (*Malkova et al., 2005*). This end can be captured by homologous sequences and used as a primer for replication (*Saini et al., 2013*). Break-induced replication is extremely processive and can extend the length of entire chromosomes. The Pif1 helicase contributes to this processivity (*Davis and Symington, 2004*; *Wilson et al., 2013*). We hypothesized that BIR might be the pathway that repaired hybrid-induced damage and that the *PIF1-E467G* allele modulated BIR.

To test this idea, we introduced *PIF1-E467G* into a previously characterized in vivo system for assessing BIR (*Anand et al., 2014*). Briefly, these strains carry an inducible HO endonuclease cut site centromere-distal to an incomplete *URA3* gene on chromosome V. Upon induction of a DSB, a telomere needs to be added to the chromosome to restore viability to the cell. This repair can proceed with BIR using homology from an incomplete *URA3* repair template on the opposite arm. The repair template is situated in a position that requires either 30 or 80 kb of BIR for telomere addition, the latter being less efficient. Repair by BIR results in a fully functional *URA3* gene, thereby conferring the ability to grow on media lacking uracil.

In a wild-type strain, the frequency of BIR using the 30 kb template was approximately 12%, consistent with previously published results (*Figure 6A*). In the absence of a functional Pif1 allele (*pif1-m2*), repair by BIR dropped to approximately 5%. Similarly, the frequency of BIR in cells carrying *PIF1-E467G* dropped to around 3%. In the 80 kb repair template strain, a similar pattern in BIR efficiency was seen (*Figure 6B*). Strains carrying the *pif1-m2* or *PIF1-E467G* alleles saw drops in BIR frequency from approximately 5% to 1%. This led us to conclude that *PIF1-E467G* inhibited BIR.

The common phenotype for *PIF1-E467G* and *pif1-m2* was inhibition of BIR, suggesting that this inhibition was responsible for allowing repair of the otherwise lethal hybrid-induced DNA damage.

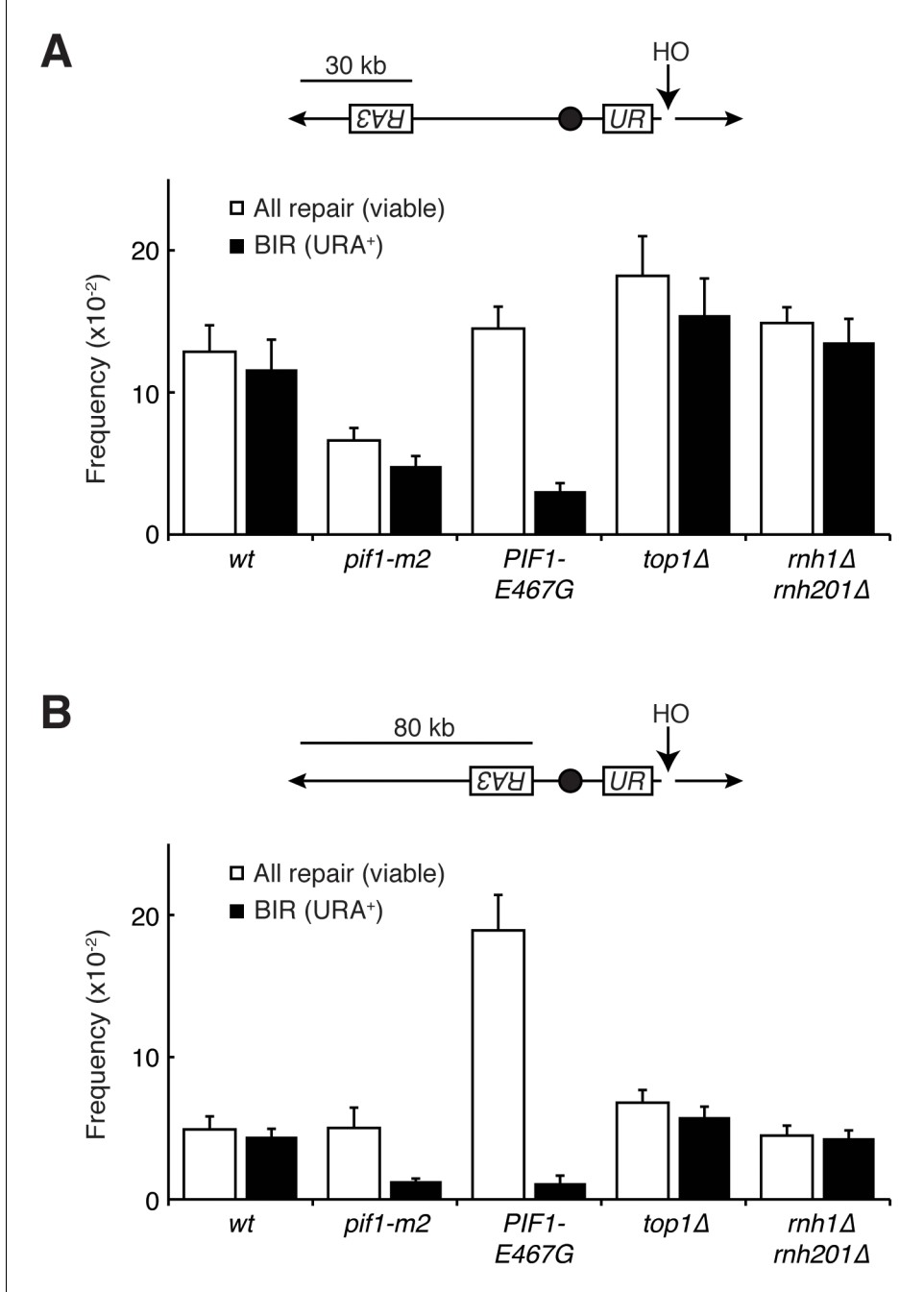

**Figure 6.** Pif1 mutants inhibit break-induced replication. (**A**) *Top:* Schematic of 30 kb repair template strain. The HO endonuclease is under control of a GAL promoter. In the presence of galactose, it is expressed, inducing a DSB on chromosome V. Sequences telomeric to the HO cut site are non-essential. Homology between the two incomplete *URA3* fragments allows for BIR and subsequent telomere addition. *Bottom:* Frequencies of repair. The percentage of cells that are viable on galactose (compared to total cells plated on YPD) indicates the frequency of all repair events. The subset of those cells that grow on media lacking uracil (URA[+]) indicates the frequency of BIR events. (**B**) As in (**A**), but with a repair template 80 kb from the telomere. Bars represent mean +/- standard deviation ($n = 4$).

The following figure supplement is available for figure 6:

**Figure supplement 1.** Mutants don't affect non-homologous end joining or HO-induced DSBs.

To test this model further, we deleted *POL32* in *rnh1Δ rnh201Δ TOP1-AID* cells (*Figure 5B*). Pol32 is a non-essential subunit of the primary BIR polymerase (Polδ), and is required for replication fork processivity (*Wilson et al., 2013*). These strains were no longer sensitive to auxin, corroborating the conclusion that inhibiting BIR was sufficient to allow for repair of otherwise lethal hybrid-induced DNA damage. Taken together, these results suggested that the inability to repair DNA damage in *rnh1Δ rnh201Δ* cells depleted of Top1 was caused by BIR processivity.

As expected, in the wild-type reporter strain, almost all DSBs were repaired by BIR (*Figure 6*). The inhibition of BIR, either by increasing the distance the repair machinery had to replicate or by inhibiting Pif1, resulted in decreased viability. In contrast, while *PIF1-E467G* cells also had reduced rates BIR, viability remained at wild-type levels (*Figure 6A and B*). Thus, BIR was compromised, but an alternative pathway stepped in to promote telomere addition. As expected if this repair was independent of BIR, this alternative pathway was as efficient with the 80 kb repair template as it was in cells with the 30 kb repair template – *PIF1-E467G* cells in the 80 kb repair strain had a total level of repair that was 5-fold greater than wild-type cells but only six percent of which was repaired using BIR. This alternative pathway was poorly activated in *pif1-m2* cells, as evidenced by a depressed level of viability. The increase in repair was not due to increased non-homologous end joining (NHEJ), as less than one percent of cells of all genotypes retained a telomeric drug resistance marker (*Figure 6—figure supplement 1A and B*). Additionally, all strains efficiently induced DSBs, as evidenced by the failure to PCR-amplify DNA surrounding the cut site after HO induction (*Figure 6—figure supplement 1C*). Further analyses will be necessary to identify the alternative pathway for repair of hybrid-induced damage.

## RNA polymerase I is a barrier to the repair of hybrid-induced DNA damage

Our genetic screen also identified *RPA190-K1482T* and *RPA190-V1486F*, two novel alleles of Rpa190, the largest subunit of RNA pol I (*Figure 7A*). These alleles were particularly intriguing because previous studies have shown that the majority of Rad52-GFP foci seen in RNase H mutants co-localize with the nucleolus. This suggested that the lethal events seen in *rnh1Δ rnh201Δ top1Δ* cells were rDNA-specific (*Stuckey et al., 2015*). Consistent with these findings, we observed efficient repair of a DSB on chromosome V by BIR and NHEJ in *top1Δ* and *rnh1Δ rnh201Δ* cells. These results suggested that topological stress and persistent R-loops were particularly problematic in the rDNA (*Figure 7* and *Figure 7—figure supplement 1*).

*RPA190-K1482T* and *RPA190-V1486F* shared the suppression phenotypes of *PIF1-E467G*. Both *RPA190* alleles suppressed the auxin-induced inviability of *rnh1Δ rnh201Δ TOP1-AID* cells (*Figure 7A*). Similarly, neither allele prevented the accumulation of high levels of foci, but rather allowed for their repair (*Figure 7C and D*, *Figure 7—figure supplement 1*). Given the specificity of RNA pol I for transcribing regions of the rDNA locus, these results strongly suggested that the lethality in *rnh1Δ rnh201Δ top1Δ* cells was due to irreparable hybrid-induced DNA damage in the ribosomal repeats, and that altering the function of RNA polymerase I allowed this damage to be repaired.

High rates of lethal damage at the ribosomal locus suggested that the structure of chromosome XII was altered. In previous studies, abnormal chromosome XII structures had been revealed by altered migration in pulsed-field gels (*Christman et al., 1993*). While chromosome XII failed to enter pulsed-field gels in *rnh1Δ rnh201Δ TOP1-AID* cells treated with auxin (*Figure 7—figure supplement 2*), chromosome XII also failed to enter pulsed-field gels in viable *top1* mutants (*Christman et al., 1993*). These results confounded the use of pulsed-field gel electrophoresis to analyze the structure of the lethal damage.

To begin to understand the mechanism by which these alleles modify the activity of RNA pol I, we turned to previously reported crystal structures of the RNA pol I complex. Residues K1482 and V1486 map to the 'jaw' domain of RNA pol I, along potential contacts with the non-essential RNA pol I subunit Rpa12 (*Figure 7—figure supplement 3*) (*Engel et al., 2013*; *Fernández-Tornero et al., 2013*). They are distinct from a previously tested Rpa190 allele that has been shown to suppress *rnh1Δ rnh201Δ top1Δ* inviability (*rpa190-3*), and which is found closer to the dNTP entry pore of the polymerase (*Stuckey et al., 2015*; *Wittekind et al., 1988*).

Rpa12 has a role in promoting RNA pol I backtracking and transcript termination (*Kuhn et al., 2007*; *Martinez-Rucobo and Cramer, 1829*). This backtracking activity may affect genome stability –

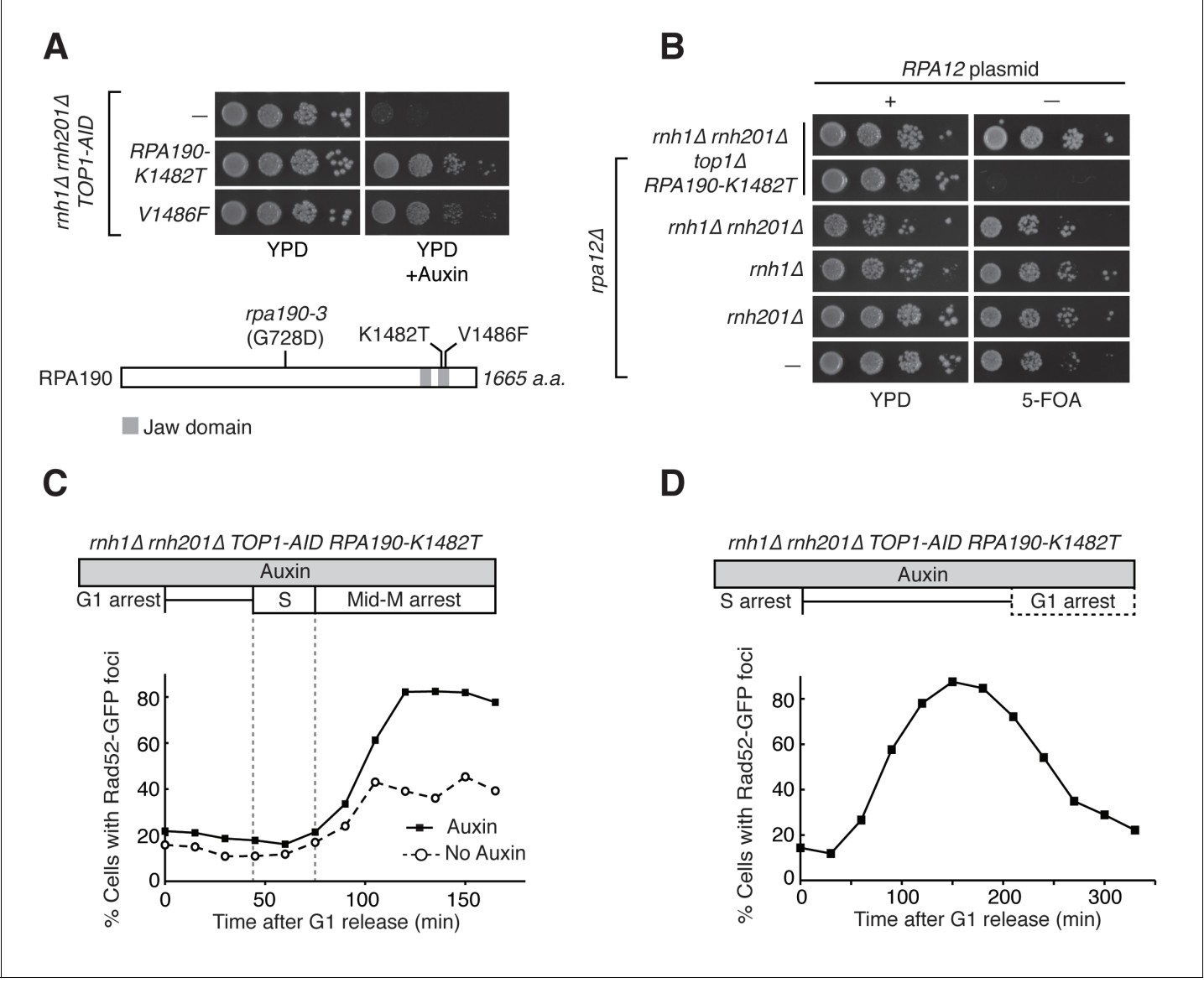

**Figure 7.** *RPA190* mutants allow for repair of R-loop induced damage. (**A**) *Top:* Rpa190-K1482T and -V1486F suppress auxin sensitivity of *rnh1Δ rnh201Δ TOP1-AID* cells. 10-fold serial dilutions of saturated cultures were plated onto YPD or YPD with auxin. *Bottom:* Schematic of Rpa190 showing the location of mutations and the jaw domain, as previously published (**Engel et al., 2013**; **Fernández-Tornero et al., 2013**). (**B**) Rpa12 is required in *rnh1Δ rnh201Δ TOP1-AID RPA190-K1482T*. Cells carrying a plasmid expressing *RPA12* and *URA3* were plated onto media lacking uracil (-URA, selects for plasmid) or media containing 5-floroorotic acid (5-FOA, selects for plasmid loss). 10-fold serial dilutions are shown. (**C**) Rpa190-K1482T does not change accumulation of Rad52-GFP foci. Experiment in *Figure 4A* was repeated on *rnh1Δ rnh201Δ TOP1-AID RPA190-K1482T* cells. (**D**) Rpa190-K1482T allows for repair of Rad52-GFP foci. Experiment in *Figure 2B* was repeated on *rnh1Δ rnh201Δ TOP1-AID RPA190-K1482T* cells in the presence of auxin.

The following figure supplements are available for figure 7:

**Figure supplement 1.** Flow cytometry on *rnh1Δ rnh201Δ TOP1-AID RPA190-K1482T* cells.

**Figure supplement 2.** Pulsed-field gel electrophoresis of R-loop mutants.

**Figure supplement 3.** Structural analysis of Rpa190 in the context of the RNA pol I complex.

studies in bacteria have shown that backtracked RNA polymerases can cause R-loop dependent DSBs due to co-directional collisions with replisomes (*Dutta et al., 2011*). The proximity of Rpa12 to our suppressor mutations suggested that Rpa12 activities might be important for repair of hybrid-induced damage. To test this idea, we knocked out Rpa12 in *rnh1Δ rnh201Δ top1Δ RPA190-K1482T* cells (*Figure 7B*). Deleting Rpa12 in these cells was lethal, indicating that our new *RPA190* alleles depended upon Rpa12. By inference, polymerase backtracking and transcript termination were required to suppress hybrid-induced lethality.

## Discussion

Our observations in this study suggest that the absence of the two RNases H leads to DNA damage that is difficult to repair after completion of S-phase. We observe an increase in Rad52 foci only in cells lacking both RNases H (*rnh1Δ rnh201Δ*). The increase in foci begins at the exit from S-phase and continues until mid-M. We show that this damage induces a significant pre-anaphase delay by activating the Rad9-dependent checkpoint. When measured in a bulk population, foci disappear slowly such that even after most cells have lost foci and completed cell division, a subset of cells retain foci and remain arrested pre-anaphase. This phenotype is indicative of an inability to efficiently repair damage. The depletion of Top1 in *rnh1Δ rnh201Δ* cells appears to exacerbate this problem by generating more foci that lead to lethal, irreparable damage and permanent arrest. Importantly, disrupting factors that modulate BIR allows for repair of these foci and restores cell division without reducing the initial level of damage. This demonstrates that the inability to repair damage, not the level of damage per se, in *rnh1Δ rnh201Δ* cells is the root cause of the inability to proceed through the cell cycle. Taken together, these results suggest that DNA:RNA hybrids inhibit DNA repair and that a critical role of the RNases H is to remove hybrids so that efficient repair can occur.

While hybrids have been recognized for many years as agents of genome instability, most studies have focused on their ability to generate DNA damage rather than their ability to alter DNA repair. However, a number of observations support our hypothesis of R-loops as inhibitors of repair. Inactivation of RNase H2 (*rnh201Δ*) by itself leads to large increases of genomic hybrids and loss of heterozygosity compared to wild-type (*Wahba et al., 2016*; *O'Connell et al., 2015*). This result suggests that loss of RNase H2 generates elevated levels of DNA damage. However, dramatically elevated Rad52 foci are only observed in *rnh1Δ rnh201Δ* cells – not *rnh201Δ* cells. We suggest that damage may be repaired rapidly in cells lacking RNase H2, while damage is repaired slowly in cells lacking both RNases H, causing foci to accumulate and persist.

Additionally, *sin3Δ* cells have elevated R-loops and hybrid-mediated genome instability. Inactivation of RNase H1 in *sin3Δ* cells increases genome instability further, but skews the events from chromosome repair to chromosome loss (*Wahba et al., 2011*). This result also supports a role for RNase H1 as critical in allowing repair of hybrid-induced damage. Presumably, under conditions of elevated hybrid formation such as *sin3Δ*, inactivation of RNase H1 alone is sufficient to cause a repair problem.

An insight into a potential role for the RNases H in DNA repair comes from a key observation in this study: lethality in *rnh1Δ rnh201Δ* cells when they are depleted of Top1 can be suppressed by mutations in Pif1 or Pol32 that inhibit BIR. The severity of the lethality – 88% cell death in a single cell cycle – suggests that BIR is a major pathway for the repair of R-loop induced damage in RNase H deficient cells. Consistent with this conclusion, previous studies mapped recombination events genome-wide in RNase H single and double mutants and found that 50% of the repair events occurred through BIR (*O'Connell et al., 2015*). Furthermore, the percent of repair by BIR was elevated about five fold in the double mutant compared to either of the singles or wild-type, although validation of this difference awaits a larger sample size.

To explain the BIR bias, we suggest that the RNases H remove hybrids from chromosomes both before and after R-loops induce DSBs (*Figure 8A*). Conversely, in the absence of RNase H activity, more DSBs are induced and hybrids persist at these DSBs. While one free end of the DSB may be properly processed by HR machinery, the presence of a hybrid on the opposite free end may block resection and/or invasion of homologous sequences. Ultimately, failure to capture the second free end of the DSB leads to BIR.

Why does hybrid-induced BIR lead to cell-cycle arrest and a complete abrogation of repair when Top1 is depleted? An important clue comes from the fact that the lethality of *rnh1Δ rnh201Δ top1Δ*

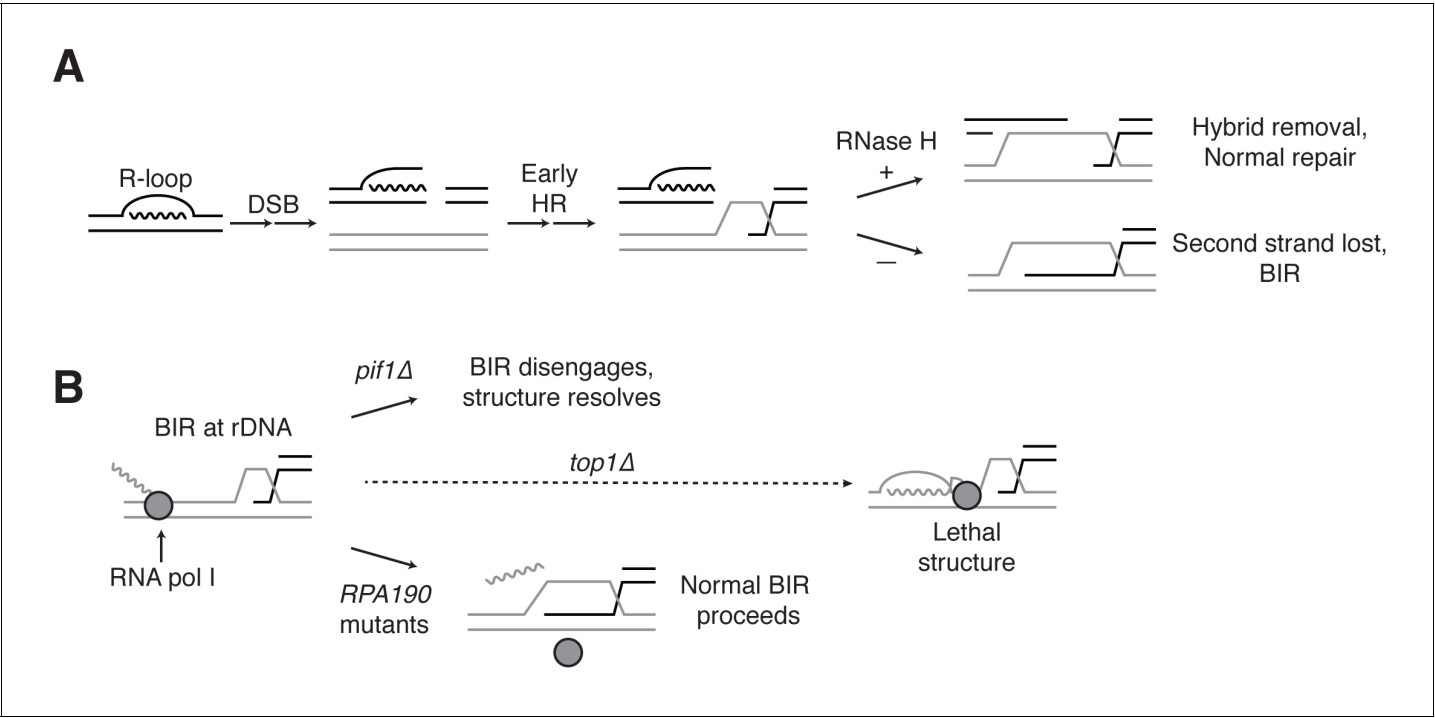

**Figure 8.** Proposed model for R-loop induced instability. (**A**) R-loops that cause DSBs persist at the break site. Early HR events (resection, homology search, strand capture and invasion) proceed as normal for one side of the break, but are inhibited by the presence of an R-loop on the opposite side. If RNase H1 or H2 act to clear the hybrid, repair can proceed as normal. If the hybrid persists, the second strand cannot be captured and the cell engages BIR. (**B**) BIR at the rDNA encounters replication blocks and slows. These replication blocks (over/under-winding, transcribing or stalled RNA polI, R-loops) are exacerbated in the absence of Top1, creating unresolvable structures that lead to cell death. *PIF1* and *POL32* mutants make BIR less processive, allowing BIR machinery to disengage before lethality occurs. The repair mechanism used after BIR is disengaged is unknown. *RPA190* mutants allow for resolution of these structures, perhaps by disengaging RNA pol I through termination or backtracking activities.

cells can be suppressed by mutations in RNA pol I, an enzyme whose function is limited to transcribing ribosomal DNA. This result strongly suggests that the lethal damage we observe is due to DNA damage in the rDNA. A number of observations corroborate this conclusion. The rDNA is the biggest source of R-loops in cells, accounting for almost 50% of all hybrids in yeast (*Wahba et al., 2016*). A previous study showed that Rad52-GFP foci co-localize with the nucleolus in *rnh1Δ rnh201Δ* cells and in *rnh1Δ rnh201Δ* cells depleted of Top1 activity (*Stuckey et al., 2015*). In addition, Top1 inactivation preferentially induces genome instability in the rDNA (*El Hage et al., 2010*; *Christman et al., 1993*, *1988*). These results suggest that BIR at the rDNA may be particularly challenging in the presence of hybrids and even more challenging in the absence of Top1. We therefore propose that the induction of hybrids in the rDNA generates a barrier to the processivity of DNA replication during BIR, thereby slowing repair (*Figure 8B*). Further inactivation of Top1 causes elevated hybrids and stalled polymerases, which we hypothesize terminally block BIR replication fork progression. This trapped BIR intermediate is an aberrant structure that then leads to lethality.

The mutations in RNA pol I that suppress the lethality of *rnh1Δ rnh201Δ top1Δ* map to the interface between subunits Rpa190 and Rpa12. This suppression is dependent upon Rpa12, a factor known to alleviate stalled polymerases either by promoting backtracking or transcription termination. Stalled RNA polymerases have been linked to R-loop dependent replication fork collisions that cause DSBs (*Dutta et al., 2011*). We therefore suggest that the *RPA190* suppressor mutations activate Rpa12, allowing it to remove stalled polymerases and possibly the associated hybrids, thereby removing the impediment to BIR imposed by Top1 (*Figure 8B*).

Our model for stalled BIR intermediates as the cause of lethality in *rnh1Δ rnh201Δ top1Δ* cells is supported by the kinetics of repair and the molecular functions of Pif1 and Pol32. BIR is known to be a multi-step process in which strand invasion happens rapidly followed by a long delay before

replication initiates (*Jain et al., 2009*). This delay would explain the slow disappearance of hybrid-induced Rad52-GFP foci in *rnh1Δ rnh201Δ* cells. After this pause, Pif1 and Pol32 are required for processivity of the BIR replication fork. We suggest that inhibition of these two factors causes the invading strand to dissociate before it can reach the blocks imposed by the hybrids and/or RNA polymerases. A slower, alternative pathway can then repair the released strand.

The identity of this alternative repair pathway remains unclear, but we speculate that single-strand annealing (SSA) may be involved. SSA is a form of HR that occurs when 5' to 3' resection uncovers complementary sequences that then anneal to each other, resulting in deletion of the intervening DNA. The high copy number of direct repeats present at the rDNA makes the locus an ideal substrate for this repair pathway. Additionally, the kinetics of end resection (~4 kb/hr) and the size of a single rDNA repeat (9.1 kb) suggest that the repair of a DSB using SSA would take around 2–3 hr per repeat (*Fishman-Lobell et al., 1992*). These estimates are consistent with the kinetics of Rad52-GFP foci dissolution seen here.

In summary, we show that the RNases H play a critical role in promoting proper repair of hybrid-induced DNA damage, particularly in highly transcribed repetitive DNA. This conclusion came from directly limiting RNase H activity in cells, but other conditions may also effectively limit RNase H. For example, many RNA biogenesis mutants induce hybrid formation and elevate genome instability (*Wahba et al., 2016*). The genome instability of these mutants can be suppressed by overexpression of RNase H, implying that RNase H activity becomes limiting when hybrid levels exceed a threshold. It has been assumed that this instability results only from increased damage induced by the persistence of R-loops. However, in light of our results, it is likely that limiting RNase H activity in these mutants also allows hybrids to interfere with repair. Finally, the particular sensitivity of the highly transcribed rDNA repeats is intriguing given that many cancer cells contain highly transcribed amplicons. These amplicons may be not only sites of R-loop formation and R-loop induced DNA damage, but also sites of improper repair.

## Materials and methods

### Yeast strains, media, and reagents

Details on strain genotypes can be found in *Supplementary file 1A*. Plasmids used in this study can be found in *Supplementary file 1B*. YPD and synthetic complete minimal media were prepared as previously described (*Guacci et al., 1997*). For all cultures and plates using auxin, a 1M stock of 3-indoleacetic acid (Sigma-Aldrich, St. Louis, MO) in DMSO was made and added to a final concentration of 500 μM. All auxin-treated experiments were compared to experiments that were mock-treated with equivalent volumes of DMSO. 5-fluorooritc acid (US Biological, Salem, MA) was used at a final concentration of 1 mg/ml.

### Dilution plating assays

Cells were grown to saturation at 30°C in YPD. Cultures were then plated in 10-fold serial dilutions. Plates were incubated at 23°C. Representative images of experiments performed in duplicate or triplicate are shown.

### Chromosome spreads

Cells were collected and spheroplasted (0.1 M potassium phosphate [pH 7.4], 1.2 M sorbitol, 0.5 mM MgCl$_2$, 20 mM DTT, 1.3 mg/ml zymolyase) at 37°C for 10 min or until > 95% of cells lysed upon contact with 1% SDS. Spheroplasting reaction was stopped by washing and resuspension in a solution containing 0.1 M MES, 1 mM EDTA, 0.5 mM MgCl$_2$, and 1M sorbitol (pH 6.4). Spheroplasts were placed onto slides and simultaneously lysed (1% Lipsol [v/v]) and fixed (4% paraformaldehyde [w/v], 3.4% sucrose [w/v]) by spreading solutions together across the slides using a glass pipette. Slides were left to dry at room temperature overnight. Indirect immunofluorescence was then performed as previously described (*Wahba et al., 2013*).

### Synchronous releases

Cells were grown to mid-log phase at 23°C in YPD. For G1 releases, alpha factor (Sigma-Aldrich) was added to 10$^{-8}$ M and cultures were incubated for approximately 3.5 hr, until >95% of cells were

visually confirmed to be arrested in G1. Cultures were split and either treated with auxin or mock treated for two hours. Cells were then collected and washed six times in 1 mL of YPD containing 0.1 mg/ml Pronase E (Sigma-Aldrich), with or without auxin, depending on treatment. Cells were then resuspended in YPD containing nocodazole (Sigma-Aldrich) at a concentration of 15 µg/ml, with or without auxin, depending on treatment. Cultures were then grown at 23°C.

For S-phase releases, cells were arrested in hydroxyurea (Sigma-Aldrich) at a final concentration of 200 mM for 3 hr at 23°C. Cultures were split and treated with auxin or mock treated for 1.5 hr. Cells were washed 6×1 mL with YPD and released into YPD containing $10^{-8}$ M alpha factor (with or without auxin, depending on treatment). Cultures were then grown at 23°C. Note that all strains used in all time courses were *bar1Δ* to allow for greater sensitivity to alpha factor.

For all Rad52-GFP foci time courses, one representative data series is shown. Experiments were performed in duplicate or triplicate, but due to subtle variation in cell cycle progression, replicates could not be combined. S-phase is notated for each individual data series based on flow cytometry. Cell cycle experiments were performed at 23°C to slow down the cell cycle relative to 30°C, thereby allowing for greater temporal resolution. Experiments performed at 30°C had identical phenotypes.

## Flow cytometry

Fixed cells were washed twice in 50 mM sodium citrate (pH 7.2), then treated with RNase A (50 mM sodium citrate [pH 7.2]; 0.25 mg/ml RNase A; 1% Tween-20 [v/v]) overnight at 37°C. Proteinase K was then added to a final concentration of 0.2 mg/ml and samples were incubated at 50°C for 2 hr. Samples were sonicated for 30s or until cells were adequately disaggregated. SYBR Green DNA I dye (Life Technologies, Carlsbad, CA) was then added at 1:20,000 dilution and samples were run on a Guava easyCyte flow cytometer (Millipore, Billerica, MA). 20,000 events were captured for each time point. Quantification was performed using FlowJo analysis software.

## Microscopy

Cells were collected and resuspended in fixative (paraformaldehyde 4% [w/v] and sucrose 3.4% [w/v]) for 15 min at room temperature followed by washing and storage in 0.1 M potassium phosphate (pH 7.4), 1.2 M sorbitol. When indicated, nuclei were visualized by brief permiabilization of fixed cells with 1% Triton X 100 (v/v) followed by staining with DAPI at final concentration of 1 µg/ml. Scoring and image acquisition was with an Axioplan2 microscope (100× objective, numerical aperture [NA] 1.40; Zeiss, Thornwood, NY) equipped with a Quantix CCD camera (Photometrics, Tucson, AZ).

## Western blotting

Western blots were performed as previously described (*Eng et al., 2015*). Primary antibodies used were a mouse monoclonal anti-V5 used at a 1:5000 dilution (Invitrogen, Carlsbad, CA) and a mouse monoclonal anti-Tub1 used at 1:20,000 dilution. Secondary antibody used was an HRP-conjugated goat anti-mouse at 1:20,000 (BioRad, Hercules, CA).

## Genetic screen

Multiple independent cultures of strain JA271a (*Mata rnh1Δ rnh201Δ top1Δ +pRS316-RNH1*) were grown to saturation in YPD to allow for loss of plasmid pRS316-RNH1 (*RNH1 CEN URA3*). Cultures were plated on media containing 5-FOA to select for loss of the plasmid. Frequency of 5-FOA resistance was approximately $10^{-7}$. Colonies were patched onto 5-FOA to confirm resistance, then confirmed to be *rnh1Δ, rnh201Δ, top1Δ*, URA⁻, *grande* (able to grow on glycerol as the sole carbon source, indicating functional mitochondria), and not carry any temperature sensitivities. Cells were grown to saturation in YPD, genomic DNA was extracted, and libraries were prepared using an IlluminaTruSeq kit (San Diego, CA). Libraries were multiplexed and sequenced with a minimum of 14-fold coverage. Sequences were mapped to an S288c reference genome and SNPs were called relative to the parental JA271a strain.

Sequencing results are summarized in *Supplementary file 2*. 25 SNPs were initially chosen for further analysis on the basis of the SNP being a nonsynonymous substitution lying within a gene with either unknown function or a known role in DNA metabolism. After PCR confirmation, several of the SNPs detected by whole-genome sequencing were found to have arisen in the outgrowth step

before genomic DNA extraction and were therefore secondary, non-causative mutations (12/25 SNPs). To test if the remaining 13 primary SNPs were causative for the phenotype, suppressor strains were crossed to strain JA290a (*Mat**alpha** rnh1Δ rnh201Δ top1Δ +pRS316-RNH1*). Diploids were sporulated and tetrads were dissected. Only four SNPs were found to consistently segregate with the *rnh1Δ rnh201Δ top1Δ* genotype in spores that had lost the pRS316-RNH1 complementation plasmid. Of these four SNPs, the two *RPA190-V1486F* hits are likely to be clonal because they arose from the same initial outgrowth culture. The three remaining suppressors – *RPA190-K1482T*, *RPA190-V1486F*, and *PIF1-E467G* – are discussed here. All three suppressors were built into *rnh1Δ rnh201Δ top1Δ* strains to confirm that they were causitive before being built into *TOP1-AID* strains.

### BIR assay

Experiments were performed as previously described (*Anand et al., 2014*). Strains were grown on YPD +cloNAT plates. Individual colonies were picked and serially diluted in water so that ~ 200 cells were plated onto YPD and ~ 2000 cells were plated on YP-GAL. Cells that grew on YP-GAL were counted then replica-plated onto SC –URA and YPD +cloNAT plates. Total survivors were calculated by dividing the number of colonies that grew on YP-GAL by the number that grew on YPD, adjusting for 10-fold dilution. Similar calculations were performed on SC –URA and YPD +cloNAT plates to determine rates of BIR and NHEJ, respectively.

### Pulsed-field gel electrophoresis (PFGE)

Cultures were grown in YPD at 23°C to saturation or mid-log. Mid-log cells were treated with auxin, nocodazole (15 μg/ml), or mock treated with DMSO for four hours. Equivalent amounts of cells were collected and subjected to PFGE and Southern blotting as previously described for large molecular weight resolution (*Hage and Houseley, 2013*).

## Acknowledgements

We thank A Zimmer, L Costantino, H Tapia, B Robison, R Lamothe, A Muir, and S Kim for comments on the manuscript. We would also like to thank VA Zakian for plasmids and JE Haber and for strains and helpful discussions. This work was funded by the National Institutes of Health (1R35GM118189-01) and in part by a National Science Foundation Graduate Research Fellowship to JA.

## Additional information

### Funding

| Funder | Grant reference number | Author |
|---|---|---|
| National Institutes of Health | 1R35GM118189-01 | Jeremy D Amon<br>Douglas Koshland |
| National Science Foundation | | Jeremy D Amon |

The funders had no role in study design, data collection and interpretation, or the decision to submit the work for publication.

### Author contributions

JDA, Conception and design, Acquisition of data, Analysis and interpretation of data, Drafting or revising the article; DK, Conception and design, Analysis and interpretation of data, Drafting or revising the article

### Author ORCIDs

Jeremy D Amon, http://orcid.org/0000-0002-8748-5228

Douglas Koshland, http://orcid.org/0000-0003-3742-6294

## Additional files

**Supplementary files**
• Supplementary file 1. Strains and plasmids used in this study

• Supplementary file 2. Details on candidate mutations in suppressor strains. Genomes of suppressor strains were sequenced and SNPs were called relative to parental strain JA271a. All non-indel alterations are listed along with the alteration conferred ('no feature' – mutation is in an intergenic sequence; 'syn.' – synonymous mutation). 'Primary?' refers to whether or not this mutation arose in the initial selection step. All primary mutations were then tested for causality. 'Causative?' refers to whether or not viable *rnh1Δ rnh201Δ top1Δ* spores from a cross with the suppressor strain consistently carried the genetic alteration.

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
