## [Decision Letter]

Thank you for submitting your article "RNase H enables efficient repair of R-loop induced DNA damage" for consideration by *eLife*. Your article has been reviewed by three peer reviewers, and the evaluation has been overseen by a Reviewing Editor and James Manley as the Senior Editor. The reviewers have opted to remain anonymous.

The reviewers have discussed the reviews with one another and the Reviewing Editor has drafted this decision to help you prepare a revised submission.

Summary:

This study reports the interesting observation that in budding yeast, RNA:DNA hybrids may promote instability not only by increasing DNA damage but also by interfering with DNA repair. Using Rad52-GFP foci as readout of chromosome breaks, it is shown that DNA damage accumulates in *rnh1 rnh201* mutants, which are defective for the two RNases H encoded by the yeast genome. It is also shown that DNA damage is further increased in the absence of topoisomerase I. Since the triple mutant *rnh1 rnh201 top1* is inviable, a conditional Top1 mutation is used in *rnh1 rnh201* mutants using an AID construct to identify suppressors. Interestingly, suppressors are a BIR-defective allele of the Pif1 helicase and a mutant of the large subunit of RNA polymerase I, the main RNAP involved in the transcription of ribosomal RNAs. Based on these and other observations, it is proposed a model in which R-loops induce the formation of chromosome breaks that cannot be properly repaired by homologous recombination, presumably because hybrids interfere with the resection of DNA ends. This would prevent the capture of the second end and would therefore promote BIR. For reasons that are not entirely clear at this stage, the BIR mediated completion of DNA replication in S/G2 would be particularly difficult at the rDNA array, which would induce mitotic defects and irreparable chromosome breaks.

Essential revisions:

A main concern relates to the question of whether the rDNA array is involved in this process. Indeed, it has been reported by the Maki lab (Ide et al., 2007, Mol Cell Biol 27, 568) that the rDNA array is extremely tolerant to replication defects. For instance, a massive depletion of active origins in the *orc2-1* mutant leads to a Rad9-dependent G2/M arrest, but deletion of the RAD9 gene rescues the viability of *orc2-1* cells at the restrictive temperature, presumably because yeast cells can tolerate the loss of a large fraction of their rDNA array. It is shown that the deletion of RAD9 in *rnh1 rnh201* TOP-AID cells does not restore viability, which argues that the incomplete replication of the rDNA array might not be the only cause of the loss of viability of the triple mutant. To address this possibility, completion of DNA replication of the chromosome XII relative to the other chromosomes should be monitored by PFGE. If their model is correct, the mobility of chromosome XII should be particularly affected, relative to other chromosomes.

A more complete discussion should be provided about the new pathway that would be acting. Disabling BIR presumably allows a more efficient and productive mechanism of repair to eventually be engaged, and this could be either canonical HR or single-strand annealing (SSA). The latter pathway would be particularly well suited to high copy number direct repeats. In particular, the potential involvement of SSA could be addressed by examining *rad51*.

The possibility that the Stuckey et al. 2015 results could be explained by stalled BIR intermediates should be tested experimentally, as it would provide a unified model to explain different results. It would be important to assay whether replication fork stalling in *rnh1rnh201top1* mutants at the rDNA is prevented by *pif1* or *rpa1-190* mutations.

To conclude that induction of hybrids in the rDNA generates a barrier to the processivity of DNA replication during BIR, it should be shown that *rnh1rnh201* mutants are deficient in the 30 or 80 kb BIR assays.

---

## [Author Response]

*Essential revisions:*

*A main concern relates to the question of whether the rDNA array is involved in this process. Indeed, it has been reported by the Maki lab (Ide et al., 2007, Mol Cell Biol 27, 568) that the rDNA array is extremely tolerant to replication defects. For instance, a massive depletion of active origins in the orc2-1 mutant leads to a Rad9-dependent G2/M arrest, but deletion of the RAD9 gene rescues the viability of orc2-1 cells at the restrictive temperature, presumably because yeast cells can tolerate the loss of a large fraction of their rDNA array. It is shown that the deletion of RAD9 in rnh1 rnh201 TOP-AID cells does not restore viability, which argues that the incomplete replication of the rDNA array might not be the only cause of the loss of viability of the triple mutant. To address this possibility, completion of DNA replication of the chromosome XII relative to the other chromosomes should be monitored by PFGE. If their model is correct, the mobility of chromosome XII should be particularly affected, relative to other chromosomes.*

There are three points we wish to address for this comment:

1) We agree with the reviewers that incomplete replication is likely not the cause of loss of viability in our strains, and we do not make this claim in our manuscript. In part because of results from our Rad9 experiments, we believe that we are seeing a DNA damage and repair phenomenon – any incomplete replication occurring at the rDNA is likely coincidental and not the direct cause of lethality.

2) We agree that the tolerance of the rDNA to various insults, not just replication defects, makes it a curious source of lethality. However, multiple lines of evidence point to the rDNA as the source of the lethal damage. We have changed the Discussion to make these multiple lines of evidence more transparent:

“Why does hybrid-induced BIR lead to cell-cycle arrest and a complete abrogation of repair when Top1 is depleted? […] This trapped BIR intermediate is an aberrant structure that then leads to lethality.”

3) We have performed PFGE experiments and have shown that the mobility of chromosome XII is affected in cells depleted of Top1. This is consistent with work done in the Fink lab showing that chromosome XII (and an ectopically located rDNA array) fails to enter a pulsed-field gel in *top1* mutants (Christman, et al., PNAS, 1993). As they note in their paper, this mobility defect is not necessarily due to incomplete replication, but can be caused by torsional stress, recombination structures, or DNA damage. We have added these experiments as a supplemental figure (Figure 7—figure supplement 2). The following short paragraph has been added to the Results section:

“High rates of lethal damage at the ribosomal locus suggested that the structure of chromosome XII was altered. […] These results confounded the use of pulsed-field gel electrophoresis to analyze the structure of the lethal damage.”

The following figure legend has been added: Figure 7—figure supplement 2.

The Materials and methods have been expanded to include the new protocols:

“Cultures were grown in YPD at 23°C to saturation or mid-log. Mid-log cells were treated with auxin, nocodazole (15 µg/ml), or mock treated with DMSO for four hours. Equivalent amounts of cells were collected and subjected to PFGE and Southern blotting as previously described for large molecular weight resolution {Hage:2013ey}.”

*A more complete discussion should be provided about the new pathway that would be acting. Disabling BIR presumably allows a more efficient and productive mechanism of repair to eventually be engaged, and this could be either canonical HR or single-strand annealing (SSA). The latter pathway would be particularly well suited to high copy number direct repeats. In particular, the potential involvement of SSA could be addressed by examining rad51.*

We also favor the SSA pathway due to the direct repeats present in the rDNA and the approximate kinetics, and have added the following paragraph to the Discussion section:

“The identity of this alternative repair pathway remains unclear, but we speculate that single-strand annealing (SSA) may be involved. […] These estimates are consistent with the kinetics of Rad52-GFP foci dissolution seen here.”

*The possibility that the Stuckey et al. 2015 results could be explained by stalled BIR intermediates should be tested experimentally, as it would provide a unified model to explain different results. It would be important to assay whether replication fork stalling in rnh1rnh201top1 mutants at the rDNA is prevented by pif1 or rpa1-190 mutations.*

After consulting with experts on 2D gel electrophoresis, we have determined that BIR forks are not likely to run with similar patterns as the canonical replication forks seen in the Stuckley, et al. 2015 paper. To our knowledge, BIR forks have not been analyzed by 2D gel electrophoresis. We have therefore removed these sentences from our manuscript. As for the stalled replication forks seen in the Stuckley, et al. paper, we expect them to act like the Rad52-GFP foci in this study. That is, in *rnh1∆ rn201∆ top1∆* cells in the presence of the *PIF1* or *RPA190* alleles, replication forks will still stall in G2-M. Given enough time, the stalled forks may eventually resolve as the cells continue to divide. Our paper is not focused on replication fork stalling but rather DNA repair, and so we feel that these experiments would answer a somewhat tangential question.

*To conclude that induction of hybrids in the rDNA generates a barrier to the processivity of DNA replication during BIR, it should be shown that rnh1rnh201 mutants are deficient in the 30 or 80 kb BIR assays.*

We have built these strains and performed the necessary experiments. However, we note that the BIR assays measure BIR efficiency on chromosome V, not at the rDNA locus on chromosome XII, where the lethal events are occurring. As such, inducing hybrids should have no effect on the BIR assay. Indeed, as shown in the updated Figure 6 and Figure 6—figure supplement 1, *rnh1∆ rnh201∆* cells are proficient for BIR on chromosome V. We have updated the text to reflect these new data and to make the distinction between the rDNA and chromosome V clearer:

“Our genetic screen also identified *RPA190-K1482T* and *RPA190-V1486F,* two novel alleles of Rpa190, the largest subunit of RNA pol I (Figure 6). […] These results suggested that topological stress and persistent R-loops were particularly problematic in the rDNA (Figure 6 and Figure 6—figure supplement 1).”